# General Practitioner’s Knowledge about Bariatric Surgery Is Associated with Referral Practice to Bariatric Surgery Centers

**DOI:** 10.3390/ijerph181910055

**Published:** 2021-09-24

**Authors:** Mara Egerer, Nicole Kuth, Alexander Koch, Sophia Marie-Therese Schmitz, Andreas Kroh, Ulf P. Neumann, Tom F. Ulmer, Patrick H. Alizai

**Affiliations:** 1Department of General-, Visceral- and Transplantation Surgery, RWTH Aachen University Hospital, Pauwelsstraße 30, 52074 Aachen, Germany; sopschmitz@ukaachen.de (S.M.-T.S.); akroh@ukaachen.de (A.K.); uneumann@ukaachen.de (U.P.N.); fulmer@ukaachen.de (T.F.U.); palizai@ukaachen.de (P.H.A.); 2Department of General Medicine, RWTH Aachen University Hospital, Pauwelsstraße 30, 52074 Aachen, Germany; nkuth@ukaachen.de; 3Department of Gastroenterology and Metabolic Disorders, RWTH Aachen University Hospital, Pauwelsstraße 30, 52074 Aachen, Germany; akoch@ukaachen.de

**Keywords:** bariatric surgery, stigmatization, obesity, general practitioners, referral practice, weight loss surgery

## Abstract

(1) Background: Patients seeking treatment for obesity and related diseases often contact general practitioners (GPs) first. The aim of this study was to evaluate GPs’ knowledge about weight loss surgery (WLS) and potential stereotypes towards obese patients. (2) Methods: For this prospective cohort study, 204 GPs in the region of the bariatric surgery center at the University Hospital Aachen were included. The participants filled out a questionnaire comprising general treatment of obese patients, stigmatization towards obese patients (1–5 points) as well as knowledge regarding WLS (1–5 points). (3) Results: The mean age of the GPs was 54 years; 41% were female. Mean score for self-reported knowledge was 3.6 points out of 5. For stigma-related items, the mean score was 3.3 points out of 5. A total of 60% of the participants recognized bariatric surgery as being useful. Knowledge about bariatric surgery significantly correlated with the number of referrals to bariatric surgery centers (*p* < 0.001). No significant correlation was found between stigma and referral to surgery (*p* = 0.057). (4) Conclusions: The more GPs subjectively know about bariatric surgery, the more often they refer patients to bariatric surgery specialists—regardless of potentially present stereotypes. Therefore, GPs should be well informed about indications and opportunities of WLS.

## 1. Introduction

General practitioners (GPs) are often the first healthcare professionals who obese patients interact with regarding information about obesity and its treatment [1]. As so-called “gate keepers” in primary health care, GPs play an important role in referral decisions concerning their patients’ health [2].

Bariatric surgery is highly effective for long-term weight loss and remission of obesity-related comorbidities, such as type 2 diabetes [3,4]. Additionally, weight loss surgery (WLS) is cost-effective regarding long-term weight loss and remission of type 2 diabetes in comparison with non-surgical interventions [3]. Obese patients undergoing bariatric surgery have a higher life expectancy than patients who obtain non-surgical weight loss therapy [5,6]. 

WLS has proved to be a safe procedure with extremely low morbidity and mortality [7,8,9]. Nevertheless, the number of bariatric surgeries performed in Germany is still low compared with other European countries [10]. Only 10.5 out of 100,000 morbidly obese patients underwent bariatric surgery in Germany, whereas 114.8 per 100,000 patients in Sweden and 86.0 out of 100,000 patients in France received a surgical procedure for their obesity [11]. One reason could be the low referral rate to bariatric surgery centers, as a previously conducted study showed that only 18% of participating German physicians stated that they referred their patients to a surgeon [1]. In comparison, 71% of participating Swedish primary care physicians stated that they referred patients who met the required criteria for WLS [12].

A potential cause is the high barrier for WLS by insurance companies as well as the GPs’ fear of surgical and postoperative complications, which influences the recommending and referral behaviour of health care professionals [13,14]. Insufficient knowledge about bariatric surgery might also be related to their referral practice [1]. Additionally, physicians are sometimes insecure in approaching obese patients concerning WLS. Lack of knowledge regarding surgical procedures, postoperative care and insurance coverage are the main reasons not to initiate these conversations [14,15,16]. 

Furthermore, stigmatization might affect GPs’ quality of care for obese patients [17]. Obese patients feel discriminated against in the health care environment twice as often as normal weight patients [18]. For example, more than half of GPs think, “having no will power” is the main reason for being extremely overweight [1]. Stigmatizing behaviour towards overweight and obese patients not only leads to discrimination, but also to harmful effects on their health care [19]. 

The aim of this study was to determine whether knowledge about WLS and stigma towards obese patients affect referral behaviour in the region of our bariatric surgery center. We hypothesized that GPs with more knowledge regarding WLS and holding fewer stereotypes refer more obese patients to bariatric surgery centers.

## 2. Materials and Methods

The report of our cross-sectional study followed STROBE Statement [20].

### 2.1. Participants

The sample consisted of GPs in the region of our bariatric surgery center at the University Hospital Aachen, Germany. We have used a dataset comprised of GPs settled in the province of North Rhein Westphalia provided by the Department of General Medicine of the RWTH Aachen University Hospital. The physicians were selected randomly and sent the questionnaire and a cover letter in paper form via mail between October 2019 and March 2020. GPs were asked to state their age and gender as well as their height and weight for the purpose of calculating the GPs’ BMI.

### 2.2. Questionnaire

The questionnaire was based on the well-established fat phobia scale [21] and a questionnaire developed by Jung et al. [1]. It consisted of sixteen questions, containing three sub items: general treatment of obese patients, stereotypes and stigmatization towards obese patients and knowledge regarding WLS. The answers were measured by a five-point-scale; one point was given for the worst and five points for the best possible answer (Appendix A).

### 2.3. General Treatment of Obese Patients

In this section, participants were asked about their frequency of BMI calculations (1 = never, 5 = every patient). Additionally, questions concerning confidence in initiating conversations with obese patients about their disease and confidence in providing advice if asked by a patient (1 = very insecure, 5 = very confident) were included. They were also asked about the frequency of providing follow-up care for patients undergoing WLS (1 = 0 patients, 5 = more than 5). 

### 2.4. Stereotypes and Stigmatization

Stigmatization towards obese patients was assessed using five modified questions based on the short form of the fat phobia scale [21]. The questions included adjectives describing obese people as lazy, having no self-control, unattractive, being self-responsible for their obesity and being insecure. The participants had to state whether they totally agreed (1 point) or totally disagreed (5 points) with the statements. A mean score was calculated of all 5 questions. GPs were also asked whether they agreed with the statement that WLS makes it ‘too easy’ for obese people to lose weight (1 = totally agree, 5 = totally disagree).

### 2.5. Weight Loss Surgery

The referral behaviour of participating physicians to bariatric surgery centers was analysed (1 = no patient, 5 = more than five). Furthermore, the participants were asked, whether they considered WLS as a useful tool for treating obesity (1 = not useful, 5 = very useful).

### 2.6. Knowledge Regarding WLS

Firstly, the frequency of provided information about the possibility of undergoing bariatric surgery (1 = no patient, 5 = more than five) was assessed. Secondly, the participating GPs were asked to state whether they were familiar with the regarded criteria for undergoing WLS (1 = unfamiliar, 5 = familiar). Questions concerning surgical procedures such as sleeve gastrectomy and Roux-en Y gastric bypass, as well as the mandatory preoperative multimodal weight loss program (1 = no knowledge, 5 = high knowledge) were posed. Mean scores regarding knowledge about WLS as well as standard deviations have been calculated using those four subitems. 

### 2.7. Data Analysis

Statistical evaluation was carried out using SPSS 27.0 software (SPSS, Chicago, IL, USA). Values are presented as mean and standard deviation. Significance was calculated using the two-sample t-test in case of normal distributions or Mann–Whitney U test. A two-sided *p* < 0.05 was considered statistically significant. Correlation was assessed by Spearman’s correlation coefficient. 

## 3. Results

A total of 651 GPs were contacted, and 204 responded (response rate = 31%). The participants’ mean age was 54 years (range 28–77 years); 41% of them were female (Table 1). The mean self-reported BMI of the participating GPs was 24.6 (SD 3.6) kg/m^2^; 36.8% were overweight and 5% were obese. A total of 12.3% of the participants did not provide their weight and height and therefore we could not calculate their BMI.

### 3.1. General Treatment of Obese Patients

A total of 38% of the participants stated that they calculate every patient’s BMI, whereas more than half of the GPs (53%) claimed to only calculate BMI if they found the patients noticeably overweight. A total of 90% expressed that they subjectively feel confident in approaching obese patients concerning their health issues and 80% affirmed that they feel subjectively confident in providing medical advice for obese patients who are seeking medical guidance. A total of 38% of the participants declared to have provided follow-up care for at least five patients who underwent WLS. Younger GPs (under 45) expressed significantly less confidence concerning patient education if contacted by them (3.90 vs. 4.38; *p* = 0.009) (Table 2). Male GPs and older participants showed significantly more provided follow-up care than female and younger GPs (3.59 vs. 3.06; *p* = 0.014 and 3.49 vs. 2.77; *p* = 0.007). 

### 3.2. Stereotypes and Stigmatization

The mean score for variables concerning stigmatization was 3.3 (SD 0.6), and almost one-third (32%) of the participants expressed that they find obese people unattractive (Table 3). A total of 24% of the GPs declared that they hold obese people accountable for their excessive weight. A total of 13% of the GPs agreed with the statement that it is ‘too easy’ for obese patients to lose weight with WLS. The question concerning attractiveness of obese people was answered significantly less stigmatizing by female GPs in comparison with male participants (3.27 vs. 2.93; *p* = 0.015). Normal weight GPs did not show higher stigmatization values than overweight GPs (3.32 vs. 3.37; *p* = 0.56). 

### 3.3. WLS and Knowledge Regarding WLS

A total of 60% of the participants agreed with the statement that bariatric surgery is a useful tool for treating obesity. The mean score for knowledge-related items was 3.6 (SD 0.8) out of 5 (Table 4). More than two-thirds (69%) of the participants informed at most two obese patients about WLS in the last six months. Two-thirds (65%) of the GPs stated that they knew the indication for obese patients to undergo WLS and 64% affirmed that they were familiar with the mandatory preoperative multimodal weight loss program. Regarding knowledge of surgical procedures, such as Roux-en-Y gastric bypass and sleeve gastrectomy, 86% claimed to know those procedures. Overweight and obese participants have educated their patients significantly more frequently on the possibility of WLS than normal weight GPs (2.69 vs. 2.14; *p* = 0.003) (Table 5). Older GPs (45 years and more) expressed significantly more familiarity concerning regarded criteria for undergoing WLS than younger GPs (3.86 vs. 3.16; *p* = 0.005). Self-reported knowledge about surgical procedures was significantly higher among male GPs (4.58 vs. 4.33; *p* = 0.036). Overall, overweight and obese GPs showed significantly more knowledge regarding WLS than normal weight GPs (3.7 vs. 3.4; *p* = 0.028). 

### 3.4. Referral Behavior

A quarter (25%) of the participating GPs had never referred any patient to a bariatric surgery center. The mean number of previously referred patients was 2.2 (SD 1.1). Knowledge about bariatric surgery significantly correlates with the number of referrals to bariatric surgery centers (r = 0.55; *p* < 0.001) (Figure 1). No significant correlation between stigma and referral to surgery was found (r = 0.135; *p* = 0.057). Referral of obese patients to bariatric surgery centers was independent from the GPs’ gender (*p* = 0.31). Overweight GPs significantly refer more patients to bariatric surgery centers than GPs with a BMI under 25 (mean referral rate 2.52 versus 2.11; *p* = 0.021). Younger GPs (under 45 years) had significantly lower referral rates than GPs aged 45 and above (1.9 vs. 2.4; *p* = 0.027).

## 4. Discussion

The World Health Organization has defined overweight as a BMI equal to or greater than 25 kg/m^2^ and obesity as a BMI equal to or greater than 30 kg/m^2^ [22]. More than half of Germany’s population is overweight and almost a quarter is morbidly obese [23]. The prevalence of obesity is increasing globally and so are related co-morbidities and healthcare costs [24,25]. 

As gatekeepers of the German healthcare system, GPs play an important role in referring patients to specialists [2]. In the case of obesity, specialist treatment comprises multimodal therapy concepts and WLS. However, one out of four GPs that participated in this study had never referred any patient to a specialized bariatric surgery center. A similar result was reported in a study by Jung et al., where one-third of the participating GPs and internists stated that they rarely recommended WLS [1]. Furthermore, a study by Foster et al. found that only a quarter of the participating physicians recommended bariatric surgery for patients who met the criteria [26]. 

One major obstacle for referral is proper knowledge about obesity and WLS. The more GPs in our study subjectively knew about WLS the higher was their referral rate, which is supported by findings of previous studies [1,27]. The mean score for knowledge (3.6 of 5) indicates an overall moderate self-reported knowledge concerning WLS. Our findings show that overweight and obese GPs had significantly higher scores concerning knowledge about WLS than normal weight GPs. A total of 87% of the GPs affirmed to know surgical procedures such as Roux-en-Y gastric bypass and sleeve gastrectomy. This is in line with another study where about 92% GPs stated that they are familiar with these surgical procedures [28]. Two-thirds of the surveyed GPs stated that they were aware of indications for WLS, comparable to the findings of a Canadian survey among family doctors [29]. 

A second obstacle might be conversation barriers as talking about obesity is a sensitive issue for both patient and GP. Encouragingly, a vast majority of the participating GPs in our study stated feeling confident initiating conversations as well as providing advice regarding treatment methods for obesity. Prior studies have also shown that GPs principally feel comfortable initiating conversations concerning weight loss treatments [16] and discussing WLS as a valuable treatment method for obesity [14]. 

Conversation with obese patients might additionally be hampered by the presence of stereotypes and stigmatization. For example, a previously conducted survey illustrated, that German GPs ranked lack of willpower and lack of exercise as one of the main causes of obesity [30]. Our study revealed the presence of stereotypes towards obese patients among the GPs with a mean score of 3.3 (out of 5). These findings are consistent with previously conducted studies, which also indicate that physicians tend to have a rather negative attitude towards overweight and obese patients [26,31]. As an example, one-third of the participating GPs in our cohort agreed with the statement, that obese people are unattractive. Additionally, our results indicate that stigmatization tendencies seem to be independent of the GPs’ BMI, as normal weight GPs did not show higher stigmatization values than overweight and obese GPs. Moreover, female participants showed significantly less stigmatization tendencies concerning attractiveness of obese patients than male physicians. Similar results were reported by Schwenke et al., with one difference to our study as they also found that older GPs showed less stigmatization towards obese patients [30].

Stigmatization towards obese people leads to physical and psychological distress and inadequate medical treatment [19,32]. People dealing with stigmatization are more likely to develop depression, anxiety, negative self-esteem, and body image dissatisfaction [33]. Likewise, a high prevalence of mental health disorders among obese patients has been reported in several studies [34,35]. Moreover, stigmatization has a negative effect on exercising motivation and eating habits [36,37]. About one-third of obese patients expressed that they cannot speak freely about their weight with doctors and 39% thought that doctors treat average weight patients in a nicer manner than obese patients [38]. Consequently, these patients tend to avoid or delay medical appointments in fear of facing humiliation and discrimination regarding their weight [39]. Some studies indicate that physicians pay patients with a higher BMI less respect, have more negative attitudes towards them, and show distancing behaviour [40,41].

Interestingly, our results implicate that referral behaviour seems to be independent of the GPs’ stereotypes of obese patients. GPs with a lower score concerning stigmatization-related items showed the same referral behaviour as GPs with less stigmatization tendencies. A previous study did not find a correlation between stigma against obese patients and number of referrals either [1]. 

However, stigma towards WLS itself is shown to be related to lower referral rates [1]. One-third of the GPs have agreed with the statement describing WLS as a ‘too easy way out of obesity’ in a previous study [1]. Similar results have been found in our sample, where 13% of the surveyed GPs concurred with the statement that WLS makes it ‘too easy’ for obese people to lose weight. Additionally, obese and overweight people who have lost weight through WLS were described as more lazy, less responsible for their weight loss, and less attractive in comparison with patients who lost their weight through exercise and diet [42,43]. One possibility to minimize stigmatization towards obese patients undergoing WLS could be to articulate that they are actively involved in the weight-loss process [44]. 

Interestingly, the overweight and obese GPs in our study showed higher referral rates to bariatric surgery centers in comparison with normal weight GPs. Comparable findings were reported by Balduf et al., implying a more positive attitude towards WLS among GPs suffering from similar weight-related symptoms [27]. Furthermore, participating GPs aged below 45 had significantly lower referral rates than GPs over 45.

Despite the exceptional importance of obesity in our modern society, the rate of GPs who are screening for obesity by calculating all patients’ BMI is still low [45,46]. In our study, more than half of the participating GPs stated that they only calculated the BMI of noticeably overweight patients. These findings are consistent with a study by Critchlow et al., where only 40% of the GPs calculated the BMI of visibly overweight patients [47]. Screening for obesity by systematic calculation of BMI by GPs might be a simple yet efficient strategy for referring patients that meet the criteria for WLS to bariatric surgery centers.

Finally, our findings display a small number of patients who have been receiving follow-up care by their GPs after undergoing WLS. This might be caused by the generally low number of performed WLS in Germany [10]. However, standardized life-long follow- up care for all post-operative patients is a crucial pillar for a successful long-term outcome of WLS. Patients who regularly attend support groups and receive education on changed dietary requirements are showing a BMI about 10% lower than those who do not receive follow-up care [48,49]. Reasons for patients not attending follow-up care regularly are experiencing shame in case of weight regain and having a relationship with their GP which is not strong enough to discuss present difficulties [50,51]. On the other hand, GPs that do not experience positive feedback of obese patients after WLS during follow-up visits might not refer other patients to WLS. 

There are some limitations concerning this study that ought to be addressed. We used a self-report questionnaire for knowledge and stigmatization, which implies a limited validity and objectivity. Moreover, questions regarding stigmatizing behaviour have been stated in an offensive manner, which might have led the GPs to answer in a more socially acceptable way. Nevertheless, the fat phobia scale has been proven to be a very reliable tool to determine stigmatization tendencies towards obese people [21]. Furthermore, our sample size is quite small and only GPs in the region of our bariatric surgery center were included, which could have led to a recruitment bias. However, the size is comparable to other similar studies and our response rate is higher than in previous studies [1,52]. Additionally, potential bias from the cohort of GPs who did not answer individual questions (12.3% in our sample regarding weight and height) should be considered. This could be due to the GPs’ potential excessive weight or underweight which participants might not want to share. Lastly, a selection bias should be taken into account as GPs with less knowledge regarding WLS and more stigmatization tendencies might have abstained from participating in this study.

## 5. Conclusions

As gatekeepers of the health care system, GPs are often first to interact with obese patients and therefore play a very important role in detecting obesity and referring obese patients to bariatric surgery centers. Our data suggests that the more GPs subjectively know about WLS, the more frequently they refer them to specialized surgical centers. Furthermore, the rate of systematic BMI calculation by GPs is still low. To decrease the global burden of disease caused by overweight and obesity, a future focus should be educational methods for GPs concerning detection and treatment methods of obesity.

## Figures and Tables

**Figure 1 ijerph-18-10055-f001:**
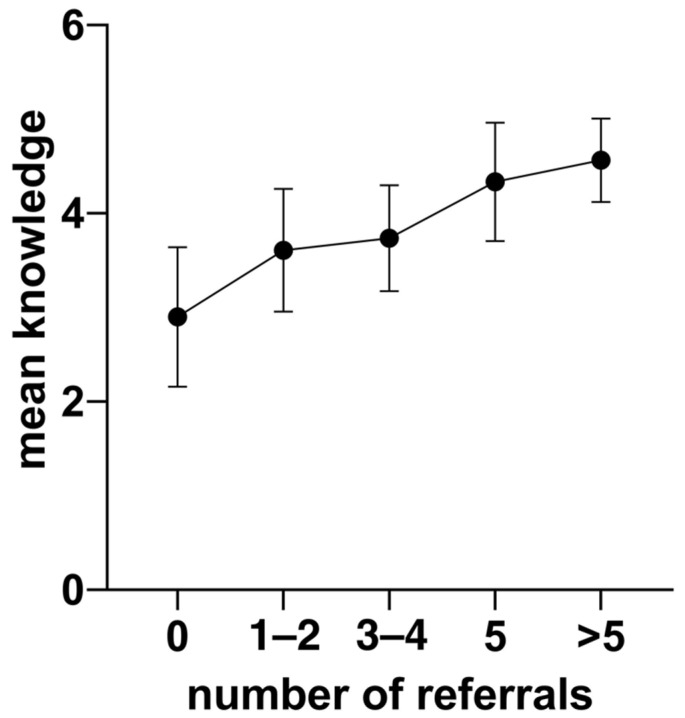
Knowledge about bariatric surgery significantly correlates with the number of referrals to bariatric surgery centers (r = 0.55; *p* < 0.001).

**Table 1 ijerph-18-10055-t001:** General characteristics of participating GPs.

	Mean ± SD	Number	Percentage [%]
Age	53.8 ± 10.4		
<45 years		32	15.7
≥45 years		144	70.6
not answered		28	13.7
Gender			
female		83	40.7
male		99	48.5
not answered		22	10.8
BMI	24.6 ± 3.6		
<25 kg/m^2^		104	50.9
≥25 kg/m^2^		75	36.8
not answered		25	12.3

**Table 2 ijerph-18-10055-t002:** General treatment of obese patients in context with GPs’ characteristics.

	Normal Weight GPs vs. Overweight/Obese GPs	Male GPs vs. Female GPs	Younger GPs vs. Older GPs
How frequently do you calculate your patients’ BMI?			
mean score ± SD	3.34 ± 0.8 vs. 3.36 ± 0.8	3.27 ± 0.9 vs. 3.45 ± 0.8	3.31 ± 1.0 vs. 3.39 ± 0.7
*p*-value	0.992	0.099	0.661
How confident are you in approaching obese patients concerning their weight and related risks?			
mean score ± SD	4.44 ± 0.7 vs. 4.47 ± 0.7	4.45 ± 0.7 vs. 4.46 ± 0.7	4.35 ± 0.8 vs. 4.48 ± 0.7
*p*-value	0.756	0.947	0.538
How confident are you in educating obese patients regarding their obesity if they ask you?			
mean score ± SD	4.22 ± 0.9 vs. 4.32 ± 0.8	4.30 ± 0.8 vs. 4.24 ± 0.9	3.90 ± 1.0 vs. 4.38 ± 0.8
*p*-value	0.549	0.783	0.009
How many patients did you provide with follow-up care after their weight loss surgery?			
mean score ± SD	3.20 ± 1.3 vs. 3.58 ± 1.4	3.59 ± 1.4 vs. 3.06 ± 1.2	2.77 ± 1.2 vs. 3.49 ± 1.3
*p*-value	0.089	0.014	0.007

**Table 3 ijerph-18-10055-t003:** Stereotypes and stigmatization.

	Mean ± SD	Percentage [%]
Overall	3.3 ± 0.6	100
Obese patients…		
are lazy	3.8 ± 1.0	
disagree (4–5)		53.0
neutral (3)		41.2
agree (1–2)		4.9
not answered		0.9
have no self-control	3.4 ± 0.9	
disagree (4–5)		41.2
neutral (3)		46.1
agree (1–2)		11.8
not answered		0.9
are unattractive	3.0 ± 1.0	
disagree (4–5)		27.5
neutral (3)		40.2
agree (1–2)		31.8
not answered		0.5
are self-responsible	3.0 ± 0.8	
disagree (4–5)		20.1
neutral (3)		55.4
agree (1–2)		24.0
not answered		0.5
have no self confidence	3.2 ± 0.9	
disagree (4–5)		33.8
neutral (3)		47.5
agree (1–2)		18.2
not answered		0.5

**Table 4 ijerph-18-10055-t004:** Knowledge regarding WLS.

	Mean ± SD	Percentage [%]
Overall	3.6 ± 0.8	100
Informing patients about WLS in the last 6 months	2.3 ± 1.1	
≥5 patients		12.3
3–4 patients		18.1
≤2 patients		69.1
not answered		0.5
Knowledge about criteria for WLS	3.7 ± 1.2	
familiar (4–5)		65.2
neutral (3)		15.3
unfamiliar (1–2)		18.6
not answered		0.9
Knowledge about pre- operative multimodal program	3.7 ± 1.1	
familiar (4–5)		63.7
neutral (3)		20.2
unfamiliar (1–2)		15.2
not answered		0.9
Knowledge about surgical procedures	4.5 ± 0.8	
familiar (4–5)		86.3
neutral (3)		9.8
unfamiliar (1–2)		3.0
not answered		0.9

**Table 5 ijerph-18-10055-t005:** Knowledge regarding WLS in context with GPs’ characteristics.

	Normal Weight GPs vs. Overweight/Obese GPs	Male GPs vs. Female GPs	Younger GPs vs. Older GPs
How many patients have you educated on the possibility of weight loss surgery in the last six months?			
mean score ± SD	2.14 ± 1.0 vs. 2.69 ± 1.3	2.43 ± 1.2 vs. 2.27 ± 1.1	2.31 ± 1.2 vs. 2.40 ± 1.2
*p*-value	0.003	0.367	0.495
Are you familiar with the regarded criteria for undergoing weight loss surgery?			
mean score ± SD	3.64 ± 1.3 vs. 3.85 ± 1.0	3.76 ± 1.2 vs. 3.65 ± 1.2	3.16 ± 1.3 vs. 3.86 ± 1.1
*p*-value	0.454	0.445	0.005
Are you familiar with the multimodal weight loss program for treating obese patients?			
mean score ± SD	3.68 ± 1.2 vs. 3.85 ± 1.0	3.77 ± 1.1 vs. 3.72 ± 1.2	3.61 ± 1.1 vs. 3.83 ± 1.1
*p*-value	0.482	0.975	0.286
Do you know surgical procedures such as Roux-en Y gastric bypass and sleeve gastrectomy?			
mean score ± SD	4.41 ± 0.8 vs. 4.59 ± 0.7	4.58 ± 0.7 vs. 4.33 ± 0.9	4.39 ± 0.8 vs. 4.55 ± 0.7
*p*-value	0.185	0.036	0.305

## Data Availability

Data available from author on request.

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
