# Peer review of "General Practitioner’s Knowledge about Bariatric Surgery Is Associated with Referral Practice to Bariatric Surgery Centers"

_ijerph, 2021, doi:10.3390/ijerph181910055_

Round 1
Reviewer 1 Report
I appreciate the authors' effort. It seems to me that the revision is satisfactory to my previous concerns. The quality of this article is now sufficient for a publication in the journal.
Author Response
Reply to Reviewers
Reviewer 1
I appreciate the authors' effort. It seems to me that the revision is satisfactory to my previous concerns. The quality of this article is now sufficient for a publication in the journal.
We would like to thank the reviewer for this kind feedback.
Reviewer 2 Report
A well written manuscript covering an interesting and important topic. The previous comments and suggestions have been addressed appropriately.
Author Response
Reply to Reviewers
Reviewer 2
A well written manuscript covering an interesting and important topic. The previous comments and suggestions have been addressed appropriately.
We would like to thank the reviewer for this kind feedback.
This manuscript is a resubmission of an earlier submission. The following is a list of the peer review reports and author responses from that submission.
Round 1
Reviewer 1 Report
This is a well written manuscript with an important and clear message. I have a few minor comments:
- The questionnaire should be included in the method section or as a supplementary appendix.
- Was the questionnaire send out in paper form or online? This might have had an effect on the return rate!
- In addition to the limitations mentioned, the likely presence of a recrution bias should be included in this section.
- Please clarify when the Pearson's correlation and when the Spearman's correlation analysis was used. To my knowledge, only the Spearman's correlation should be used for categorical data as analyzed in this study.
- A graphical presentation of (correlation plot) for the most important result (knowledge - referrals) should be included in the result section.
Author Response
General practitioner’s expertise about bariatric surgery is associated with referral practice to bariatric surgery centers
Reply to the reviewers’ comments:
The authors would like to thank the reviewers for their valuable comments. We have responded to the comments and have revised the manuscript in light of them. Details of our replies are shown below.
Reviewer 1:
Comment 1: The questionnaire should be included in the method section or as a supplementary appendix.
We have attached the questionnaire as a supplementary appendix.
Comment 2: Was the questionnaire send out in paper form or online? This might have had an effect on the return rate!
The questionnaire was sent out in paper form. We added this information to the methods.
Comment 3: In addition to the limitations mentioned, the likely presence of a recrution bias should be included in this section.
We added the potential presence of a recruitment bias to the limitations.
Comment 4: Please clarify when the Pearson's correlation and when the Spearman's correlation analysis was used. To my knowledge, only the Spearman's correlation should be used for categorical data as analyzed in this study.
We thank the reviewer for this valuable comment. Correlation was only assessed by Spearman’s correlation coefficient. This is now clarified in the methods.
Comment 5: A graphical presentation of (correlation plot) for the most important result (knowledge - referrals) should be included in the result section.
A correlation plot was added (Fig. 1).

Reviewer 2 Report
This study aimed to determine whether knowledge and stigma towards obesity affect GPs' referral behavior for WLS. The study subject was interesting, the aim was clear, and the article was very well written. I appreciate the work of the authors.
There were a few concerns and suggestions I would like to give the authors:
Introduction
- I felt like there could be more information included in the Introduction. For example, the authors mentioned that the referral rate was lower in Germany than in other European countries. What were the numbers reported in previous studies?
- The length and volume of information were not very balanced between different paragraphs. The authors might want to break the second paragraph, addressing WLS as well as its referral in one paragraph and possible barriers of referral in another one.
- The hypothesis of the study was missing in the Introduction.
Methods and Results
I felt like the authors could provide more information for this interesting topic to improve the scientific soundness of the study. There were only two tables which were quite simple and contained not much data. All the data displayed were descriptive statistics was done. While IQR, t-test, Mann-Whitney U test, and Pearson's correlation were all stated in "2.6 Data analysis", I didn't see any of them in the actual results. The study had to perform further analysis, at least to examine the correlations and associations of the dependent and independent variables, or to examine the inter-group differences in referral between GPs with different characters.
Discussion and Conclusions
The Discussion and Conclusions were very well written. However, if the authors could conduct some more in-depth analysis from the data obtained, the findings of this study would be more informative and novel to the readers.
Author Response
General practitioner’s expertise about bariatric surgery is associated with referral practice to bariatric surgery centers
Reply to the reviewers’ comments:
The authors would like to thank the reviewers for their valuable comments. We have responded to the comments and have revised the manuscript in light of them. Details of our replies are shown below.
Reviewer 2:
Comment 1: I felt like there could be more information included in the Introduction. For example, the authors mentioned that the referral rate was lower in Germany than in other European countries. What were the numbers reported in previous studies?
Data concerning referral numbers in other European countries was added (Angrisani, L.; Santonicola, A.; Iovino, P.; Ramos, A.; Shikora, S.; Kow, L. Bariatric Surgery Survey 2018: Similarities and Disparities Among the 5 IFSO Chapters. Obes Surg 2021, doi:10.1007/s11695-020-05207-7). The following sentences were included: For example, only 15.186 bariatric procedures were performed in the year 2018 in Germany, whereas 46.654 bariatric interventions were performed during the same time period in France [10].
Comment 2: The length and volume of information were not very balanced between different paragraphs. The authors might want to break the second paragraph, addressing WLS as well as its referral in one paragraph and possible barriers of referral in another one.
We thank the reviewer for this valuable comment . We have divided the second paragraph in separate paragraphs.
Comment 3: The hypothesis of the study was missing in the Introduction.
We have added the hypothesis of the study in our Introduction. We added the following sentence: We hypothesized that GPs with more knowledge regarding WLS and less stereotypes refer more obese patients to bariatric surgery centers.
Comment 4: I felt like the authors could provide more information for this interesting topic to improve the scientific soundness of the study. There were only two tables which were quite simple and contained not much data. All the data displayed were descriptive statistics was done. While IQR, t-test, Mann-Whitney U test, and Pearson's correlation were all stated in "2.6 Data analysis", I didn't see any of them in the actual results. The study had to perform further analysis, at least to examine the correlations and associations of the dependent and independent variables, or to examine the inter-group differences in referral between GPs with different characters.
We thank the reviewer for this valuable comment. We have included more results, for example, that overweight GPs showed significantly more knowledge regarding WLS than normal weight GPs.
Additionally a correlation plot regarding knowledge and referral rate was added.
Comment 5: The Discussion and Conclusions were very well written. However, if the authors could conduct some more in-depth analysis from the data obtained, the findings of this study would be more informative and novel to the readers.
We thank the reviewer for this valuable comment, changes have been made accordingly.
Normal weight GPs did not show higher stigmatization values than overweight GPs (3.32 vs. 3.37; p = 0.56).
Overweight and obese GPs showed significantly more knowledge regarding WLS than normal weight GPs (3.7 vs. 3.4; p = 0,028).
Younger GPs (under 45 years) had significantly lower referral rates than GPs over 45 (1.9 vs. 2.4; p = 0.27).

Round 2
Reviewer 2 Report
I appreciate the authors' effort on revision. However, the research depth and scientific soundness still have quite some room for improvement.
Comment 1: Please include more information on the referral rate of Germany and that of other countries, and for a fair comparison, preferably presented with the estimated total population of individuals indicated for WLS in the respective country.
Comment 2: The Introduction is now clear and easy to read with a research hypothesis added. However, paragraphs in the Discussion are still lengthy and difficult to read.
Comment 3: The results now contain more information regarding GP's demographic characters with other variables. Overweight/ obese GPs had more knowledge on WLS. GPs with more knowledge on WLS had higher referral rate. Younger GPs had lower referral rate. Would weight status and age then be the confounders in the WLS knowledge - referral relationship? There would be a threat to the internal validity of the main finding of this study then.
Author Response
General practitioner’s expertise about bariatric surgery is associated with referral practice to bariatric surgery centers
Reply to the reviewers’ comments:
The authors would like to thank the reviewer for the valuable comments. We have responded to the comments and have revised the manuscript in light of them. Details of our replies are shown below.
Reviewer 2:
Comment 1: Please include more information on the referral rate of Germany and that of other countries, and for a fair comparison, preferably present with the estimated total population of individuals indicated for WLS in the respective country.
We added information concerning referral rates and patients eligible for WLS. The following sentences were included: Only 10,5 out of 100.000 morbidly obese patients underwent bariatric surgery in Germany, whereas 114,8 per 100.000 patients in Sweden and 86,0 out of 100.000 patients in France received a surgical procedure for their obesity [11]. One reason could be the low referral rate to bariatric surgery centers, as a previously conducted study showed that only 18% of participating German physicians stated to refer their patients to a surgeon [1]. In comparison, 71% of participating Swedish primary care physicians stated to refer patients who met the required criteria for WLS [12].
Comment 2: The Introduction is now clear and easy to read with a research hypothesis added. However, paragraphs in the Discussion are still lengthy and difficult to read.
Paragraphs in the discussion were divided and some sentences were simplified. We hope that the discussion is now easier to read.
Comment 3: The results now contain more information regarding GP’s demographic characters with other variables. Overweight/obese GPs had more knowledge on WLS. GPs with more knowledge on WLS had higher referral rate. Younger GPs had lower referral rate. Would weight status and age then be the confounders in the WLS knowledge- referral relationship? There would be a threat to the internal validity of the main findings of this study then.
We thank the reviewer for this valuable comment. BMI and age of the GPs could be confounders. However, we performed a linear regression analysis with SPSS and found no statistical significance.
